# TCR Recognition of Peptide–MHC-I: Rule Makers and Breakers

**DOI:** 10.3390/ijms22010068

**Published:** 2020-12-23

**Authors:** Christopher Szeto, Christian A. Lobos, Andrea T. Nguyen, Stephanie Gras

**Affiliations:** 1Department of Biochemistry and Molecular Biology, Biomedicine Discovery Institute, Monash University, Clayton, VIC 3800, Australia; chris.szeto@monash.edu (C.S.); christian.lobos@monash.edu (C.A.L.); andrea.nguyen@monash.edu (A.T.N.); 2Australian Research Council Centre of Excellence for Advanced Molecular Imaging, Monash University, Clayton, VIC 3800, Australia

**Keywords:** human leukocyte antigen (HLA), MHC class I, peptide antigens, TCR binding, αβ TCR, δβ TCR, γδ TCR

## Abstract

T cells are a critical part of the adaptive immune system that are able to distinguish between healthy and unhealthy cells. Upon recognition of protein fragments (peptides), activated T cells will contribute to the immune response and help clear infection. The major histocompatibility complex (MHC) molecules, or human leukocyte antigens (HLA) in humans, bind these peptides to present them to T cells that recognise them with their surface T cell receptors (TCR). This recognition event is the first step that leads to T cell activation, and in turn can dictate disease outcomes. The visualisation of TCR interaction with pMHC using structural biology has been crucial in understanding this key event, unravelling the parameters that drive this interaction and their impact on the immune response. The last five years has been the most productive within the field, wherein half of current unique TCR–pMHC-I structures to date were determined within this time. Here, we review the new insights learned from these recent TCR–pMHC-I structures and their impact on T cell activation.

## 1. Overview of Structures and Status

T cells use their surface receptor, called T cell receptor or TCR, to recognise peptides presented by major histocompatibility complex (MHC) molecules. The peptide presented by MHC can be derived from the host proteins (self-peptides), pathogens (virus and bacteria), or tumours. The T cells need to differentiate between self and foreign peptides (including modify self), and are only activated upon MHC presentation of foreign epitopes. This recognition, driven by the TCR, is the critical first step of T cell activation preceding the immune response.

MHC molecules, also called human leukocyte antigens or HLA in humans, are extremely polymorphic, ensuring that a wide range of peptides can be presented to T cells. MHC molecules are divided into two main classes, i.e., I and II, and are restricted to either CD8+ or CD4+ T cells, respectively. In this review, we focus on the TCR recognition of peptide (p) presented by MHC class I (MHC-I) molecules only. The MHC-I antigen binding cleft is closed at the N-terminal and C-terminal regions (Figure 1A), while MHC-II molecules have an open-ended cleft. These differences between open and closed ends of the cleft changes the preferred length of the bound peptide, whereby MHC-I often binds shorter (8–10 residues) peptides than MHC-II (>11 residues), albeit with some exceptions [1]. MHC-I molecules have a series of pockets in the cleft, A to F, harbouring different chemical properties between different allomorphs resulting in the ability to bind different peptide repertoires. The B and F pockets are where the primary anchor residues, the second and last positions of the peptide (P2 and PΩ, respectively), bind to the MHC-I (Figure 1B). These residues are often conserved between different peptides binding to the same MHC, for example, for peptides binding to HLA-B*35:01, a P2-Pro is often observed as well as PΩ-Tyr [2,3].

To recognise the highly polymorphic MHC and the large repertoire of peptides, TCRs also need to be extremely diverse. Diversity in the TCR repertoire is achieved through the random genetic rearrangement of Variable (V or TRV), Diversity (D), and Joining (J) gene segments for TCR β chain (V and J for TCR α chain) [4]. TCRs possess three regions of variability, commonly known as the complementary determining regions (CDR1, CDR2, and CDR3), which make up the antigen-binding site that directly interacts with pMHC. The CDR1 and CDR2 regions are germline-encoded by the V gene segment, whilst CDR3 hyper-variability is further accentuated through the addition or removal of nucleotides (N) at V(D)J junctions [5]. Theoretically, the random rearrangement of TCR gene segments can give rise to a diversity of 10^15^–10^20^ T cell clonotypes [6]; however, thymic selection events during T cell maturation trims this down to a TCR repertoire of 2.5 × 10^7^ within an individual [7]. This provides enormous T cell diversity to recognise antigenic peptides presented by MHC molecules, making each TCR–pMHC complex structure unique in the recognition of pathogens by the adaptive immune system.

The first TCR–pMHC-I structure solved was by Garboczi and colleagues using X-ray crystallography in 1996 for HLA-A*02:01 presenting the Tax peptide from human T cell lymphotropic virus in complex with the A6 TCR (Table 1) [8]. Since then, 81 unique TCR–pMHC-I complexes have been solved and are available in the Protein Data Bank (PDB; Table 1), with more than half of these complexes solved in the last 5 years (Figure 1C). Despite this increase in number over the last few years, those 81 TCR–pMHC-I complexes still represent a very narrow slice of the peptide repertoire. For example, thus far only 20 different MHC allomorphs have been crystallised in complex with a TCR (Figure 1D, Table 1). The addition of these recent structures allows us to observe how TCRs can recognise HLA-A*01:01 [9]; HLA-A*11:01 [10]; HLA-B*07:02 [11]; HLA-B*37:01 [12]; HLA-A*02:06 [13]; and, for the first time, HLA-C [14].

Some of the most striking features revealed over the last few years include the first reversed docking MHC-I restricted TCR [15], how a TCR can flip the peptide upon binding [16], how δβ and γδ TCRs recognise pMHC [17,18,19], and the revelation of new rules defining TCR cross-reactivity [20].

## 2. TCR–pMHC-I Structures: Making Some Common Rules

### 2.1. Canonical TCR Docking

Until 2016, all MHC-I-restricted TCRs adopted a conserved and canonical docking topology onto the pMHC-I complex (Figure 2A). The TCRα chain was atop the MHC-I α2-helix while the TCRβ chain was above the α1-helix of the MHC-I. A notable exception to this common docking mode was revealed in a mouse model of influenza infection (Figure 2B), and is described below (Section 3, [15]). With the exception of this mouse TCR, the other 80 unique complexes of TCR–pMHC-I all exhibit conserved canonical binding (Figure 2C).

### 2.2. CDR Loops Role in Binding pMHC

The canonical binding of TCR with pMHC complex has also led to the generalisation of the role of the CDR loops, due to their most frequent localisation. Thereby, it is generally accepted, and often observed, that the germline encoded CDR1 and CDR2 loops are mainly focused on the recognition of the MHC molecule itself, while the hypervariable CDR3 loops are the main drivers of peptide recognition. This observation suggests that there might be a co-evolution of TCR genes (CDR1/2 germline encoded loops) with MHC molecules [21,22].

### 2.3. Co-Contribution of Both TCR Chains

Despite the diversity of TCR docking angle and mode observed among those 81 structures (Figure 2C), co-contribution of both TCR chains is a recurring feature. The TCRα chain contributes between 28 and 78%, with an average of 52%, while the TCRβ chain contributes to 22–72%, with an average of 48%, to the pMHC interaction (Table 1). Half of the complexes fall within the 40–60% contribution between the two chains, showing a roughly shared and balanced contribution of both TCR chains.

### 2.4. Co-Binding of Peptide and MHC

In addition to the co-contribution of both TCR chains, currently, all TCR–pMHC-I structures show that the TCR contacts both the peptide antigen and MHC. The peptide, despite its small size compared to the MHC molecule, can have a large contribution ranging from 12 to 49% of the pMHC buried surface area (BSA), with an average of 29% (Table 1). The contribution of the peptide antigen towards the pMHC–TCR interaction is a unique and shared feature among all peptide-specific T cells, for both MHC-I or MHC-II. This feature is not shared with lipid- or metabolite-derived specific TCRs for which the recognition of both MHC/MHC-like molecules and the bound antigen is not a requirement. For example, the structure of the autoreactive BK6 TCR, able to recognise multiple self-lipid antigens, solved in complex with the MHC-like molecule CD1a showed that the TCR could bind to CD1a without contacting the bound lipid [23]. This docking mechanism favours the recognition of multiple lipid antigens, decreasing TCR specificity that can in some cases lead to T cell autoimmunity. Other examples involve mucosal-associated invariant T cell (MAIT) specific for the MHC-like molecule, MR1. The MR1 molecule presents small metabolites deeply buried in the cleft that is not contacted by the MAIT TCR upon binding [24,25].

### 2.5. “On the Top” Binding Mode

Despite the large diversity of docking angles, the structures solved thus far show that there is a requirement for MHC-I restricted TCR to “sit” on the top of the cleft (Table 1). This conserved localisation of the TCR ensures that the peptide and the MHC-I helices are always contacted by the TCR. This is a specific feature of peptide-MHC-I recognition, for which thus far no exception has been observed (Figure 2C). Given that a larger number of TCR antigen structures have been solved with pMHC-I complexes, this conservation is remarkable. This “on the top” interaction is not a requirement for γδ TCR specific for the MHC-like molecule, MR1. Indeed, the recent structure of the G7 γδ TCR in complex with MR1 shows that TCR does not always need to contact the antigen-binding cleft, and that γδ TCR can adopt an “on the side” binding mode [26]. Interestingly, binding “on the top”, rather than sideways, is a common characteristic of αβ TCRs thus far. This conserved feature might be underpinned by the requirement for T cells to engage with pMHC complexes to pass thymic selection. TCR binding on top of the cleft also ensures that the co-factor molecule (CD8 and CD4) can be productively engaged with the pMHC complex, which has implication for T cell signalling and activation, and in turn, dictating the immune response.

## 3. The Recent TCR–pMHC-I Structures Are Rule Breakers

From the first structures of TCR binding to pMHC-I, some general rules were drawn that were observed in subsequent TCR–pMHC structures. However, as we are learning more about TCR interaction with pMHC complex, with twice more structures solved in the last 5 years, some exceptions have started to emerge against those earlier established rules.

### 3.1. Reverse Dockers

In 2016, we identified the basis of mice influenza-specific TRBV17+ CD8+ T cells’ poor recruitment from naïve to immune repertoire [15]. X-ray crystallography revealed that such T cells exhibit TCRs that recognised pMHC with a reversed-polarity docking topology (Figure 2B). Two of these TRBV17+ TCRs, NP1-B17 and NP2-B17, were observed to dock onto pMHC-I with 180° reversed polarity, where the TCRα chain is docked over the α1-helix and the TCRβ chain is contacting the α2-helix (Figure 2B). Interestingly, these TCRs were able to bind to pMHC-I with moderate affinities (K_d_ of 30–40 µM) relative to the range in other TCR–pMHC-I (Table 1); however, they were poorly activated upon pMHC-I recognition. This highlights that TCR–pMHC-I affinity is not the sole determinant behind T cell activation. The reversed docking topology could limit or impair CD8 co-factor binding, and the consequences for a lack of mediated downstream effects that lead to poor T cell activation. Additionally, the existence of a reversed-polarity docking TCR, poorly recruited into the immune repertoire, favours the selection model theory [27,28] in which MHC restriction may not be an intrinsic feature of TCRs through co-evolution (germline-encoded model [21,22]), but instead due to the regulatory processes within thymic selection. This suggests that the canonical docking topology observed for all αβTCRs derived from the immune pool might be a direct requirement of T cell activation.

### 3.2. CDR3 Loops Do Not Always “See” the Peptide

In addition to binding in a reversed orientation, the NP1-B17 and NP2-B17 TCRs docked with their TCRα chain shifted towards the C-terminus of the pMHC-I and perched higher than the α1-helix, with the CDR3α loop interacting with residues (MHC residues 18 and 89) outside the antigen-binding cleft (Figure 3A) [15]. The shifted TCRα chain only contacted the MHC molecule, and not the peptide at all. In addition, the reverse docking topology placed the TCRβ chain above the peptide, where the CDR2β loop and framework (FW) residues from the β-chain interacted directly with the bound peptide (Figure 3B). The CDR3β, on the other hand, formed interactions with residues within the α1-helix and not the peptide. Surprisingly, this meant that peptide interaction was solely mediated by the germline encoded CDR2β and FWβ, which forms the basis of TRBV17 gene usage.

### 3.3. When the Peptide Drives the Show

In most TCR–pMHC-I structures, the CDR1/2 loop has a larger role in contacting the MHC than the peptide. These recurrent observations lead to the theory that the germline-encoded regions of TCRs might have co-evolved to recognise MHC molecules [22]. This is aligned with the fact that the hypervariable non-germline encoded CDR3 loops are most often seen to dominate the interaction with the peptide. However, some exceptions have emerged, whereby bias usage of specific germline TRBV genes were driven by direct interaction with the peptide [10,29]. The first example in mouse [29] shows an almost exclusive use of Vβ8.1 gene (TRBV13-3) to recognise the malaria-derived SQL peptide presented by H-2D^b^. The structure of a TRBV13-3+ TCR in complex with H-2D^b^-SQL revealed that the CDR1β loop motif ^30^DY^31^ was interacting with the peptide. The Asp30β only contacted the peptide while the Tyr31β contacted both the peptide and H-2D^b^ molecule (Figure 3C). The ^30^DY^31^ motif is unique to TRBV13-3 within the mouse TRBV. Mutagenesis studies demonstrated the critical role of the ^30^DY^31^ motif in binding to peptide residues P7-Lys and P8-Tyr, showing that the TRBV13-3 bias gene usage was driven by the SQL peptide itself [29]. A second example was based on the discovery of a TRBV bias in Dengue-specific T cells [10]. We showed that TRBV11-2 was largely used in T cells and was able to recognise the NS3_133_ peptide presented by HLA-A*11:01 when the peptide was derived from three different Dengue serotypes (DENV1, DENV3, and DENV4, (GTSGSPI^I^/_V_NR)). However, if the NS3_133_ peptide was derived from DENV2 serotype (GTSGSPIIDK), the TRBV11-2 gene was avoided in the T cell repertoire. Structural analysis of a TRBV11-2+ TCR, D30 TCR, in complex with HLA-A*11:01 presenting the DENV1 and DENV3/4 derived peptides showed direct interaction between the germline-encoded CDR2β loop and the NS3_133_ peptides. Mutagenesis analysis showed the importance of the TCR-peptide interaction (Asn58β with P9-Asn). In addition, TCR sequence analysis revealed a shared CDR2β loop motif with other TRBVs able to recognise the same Dengue serotype [10].

These studies show that the peptide can be a key driver of TCR bias that is selective for specific CDR1/2 residues, which interacts directly with the peptide antigen independent of MHC restriction.

### 3.4. Does Length Really Matter?

MHC-I molecules usually present short peptides of 8 to 10 residues in length. However, advances in mass spectrometry and peptide elution analysis has revealed that longer peptides (>10 residues) could be presented by MHC-I molecules [1]. Despite this, it is unclear if a TCR is able to engage with peptides of multiple lengths or whether there is a length restriction. A study by Ekeruche-Makinde and colleagues showed that peptide length is linked with TCR engagement outcome [30]. Using a combinatorial peptide library with varied length, they showed that TCR cross-reactivity was dependent on the length of the presented peptide, and that TCRs were unable to react to peptides with different lengths [30]. Therefore, the observation from Riley and colleagues defining the basis for the DMF5 TCR cross-reactivity was surprising [16]. The DMF5 TCR is able to recognise a 10mer MART-1 epitope (ELA), can cross-react onto a 10mer SML peptide by a conserved docking mode, and is also able to cross-react with a 10mer MMW peptide presented by HLA-A*02:01 molecule. The MMW peptide is presented in a canonical fashion by HLA molecules, whereby the peptide is primarily anchored by its P2-Met and P10-Met (PΩ). Upon DMF5 TCR binding, the 10mer peptide is “flipped out” of the C-terminal region (Figure 3D), where P10-Met is pushed out of the cleft, and P9-Met becomes the primary anchor [31]. This change of conformation causes the MMW peptide to adopt a 9mer conformation in the cleft after DMF5 TCR binding. The study also showed that the MDF5 TCR could recognise a 9mer truncated version of the MMW, but not a 10mer MMW peptide with a P10-Val mutation [16]. The Val is a more favoured PΩ anchor for HLA-A*02:01 molecule and might prevent the structural change required for DMF5 TCR recognition of the 10mer MMW peptide. Altogether, this shows that the DMF5 TCR is able to engage with the 9mer peptide, even when presented with a 10mer peptide, and can recognise different peptide lengths, dramatically increasing its peptide repertoire.

## 4. The Scarlet pMHC: Multiple TCRs Recognising the Same pMHC-I

It is uncertain as to whether having multiple TCRs engaging the same pMHC complex is considered to be advantageous or detrimental for an effective functional adaptive immune response in an individual. At the population level, there is an advantage of having multiple TCRs able to engage with the same peptide, as this would provide different TCR–peptide interactions that could limit viral escape. There are a number of examples of multiple TCRs recognising the same pMHC-I (Table 1). This gives an opportunity to identify common or divergent structural features between TCRs engaging the same pMHC-I.

### 4.1. Conserved Peptide Interactions

The human cytomegalovirus (CMV)-derived NLV peptide (NLVPMVATV) is presented by HLA-A*02:01 in a canonical fashion with solvent-exposed P4-Pro, P5-Met, and P8-Thr and potential TCR interaction [32]. To date, the structures of three TCRs (RA14, C7, C25) in complex with HLA-A*02:01-NLV are available [32,33]. The TCRs dock onto HLA-A*02:01-NLV with different angles (35°, 29°, and 61°), but all docking modes bury 85–90% of the peptide solvent accessible surface [32,33]. All three TCRs contact the peptide in the same way, with the CDR1α and CDR3α loops interacting with the N-terminal part of the peptide, and the CDR3β loop contacting the C-terminal part of the peptide with the CDR1β loop contributing to these interactions for the RA14 TCR [32,33]. Interestingly, despite the additional contribution of the CDR1β, RA14 TCR has the lowest binding affinity to HLA-A*02:01-NLV when compared to C7 and C25 (Table 1). While all the TCRs interact with the peptide in a similar fashion, each uses different strategies to engage the HLA-A*02:01, which could be due to their different TCR gene usages and might underpin the different affinities observed.

### 4.2. Peg Notch or Peg Not?

The immunodominant influenza-derived M1 peptide (GILGFVFTL) presented by HLA-A*02:01 is featureless, lying flat within the HLA cleft, and has been described as “vanilla” because of this lacklustre trait [34]. Due to the lack of prominent features of this epitope, M1-specific TCRs are extremely biased with the expression of TRAV27/TRBV19 combinations [34], and the preferential selection of an RS motif within the CDR3β loop. The structure of a prototypical TRAV27/TRBV19 TCR expressed in unrelated individuals (public TCR), JM22, in complex with HLA-A*02:01-M1, showcased the first description of a shape complementarity feature described as a “peg” and “notch” interaction [34]. This interaction portrays a gap between P5-Phe and P7-Phe of the peptide, and the HLA α2-helix (Gln155 and Ala150) forms a “notch”, with the conserved Arg98 of the CDR3β forming the “peg” docks into the notch, which was found to also be able to form a network of hydrogen bonds (Figure 4A).

In addition, three more TCR-HLA-A*02:01-M1 structures have been solved (Table 1), two with TRBV19+ TCRs and one with a TRBV27+ TCR, none of them sharing the ^98^RS^99^ motif in their CDR3β loop. LS01 TCR (TRAV24/TRBV17) has a Phe instead of the Arg in its CDR3β loop (Figure 4B), which acts as the peg to form a peg–notch interaction but lacks the hydrogen bond network observed with the JM22 TCR [35]. The resulting affinity of LS01 is therefore five times lower than that of JM22 (Table 1) [35]. The LS10 TCR (TRAV38-2/DV8*01/TRBV19) structure with HLA-A*02:01-M1 shows that the P5-Phe of the peptide adopted a different conformation away from the HLA α2-helix due to interaction with the CDR3α loop. Due to the peptide structural changes, the side chain of P5-Phe fills the “notch” and instead forms a small hydrophobic pocket that interacts with the CDR3α loop (Ala98 and Gly99; Figure 4C). Despite the loss of the peg–notch interaction, the LS10 TCR still maintains an affinity similar to LS01 TCR [35], but lower than the JM22 TCR (Table 1). The F50 TCR (TRAV13-1/TRBV27) utilises a large Trp99β within its CDR3β loop to occupy the notch and contacts the HLA (Ala150, Val152, and Gln155; Figure 4D). Despite conserving a peg–notch interaction, the affinity of the F50 TCR for HLA-A*02:01-M1 is considerably lower than the other TCRs (K_d_ of 76μM) [36]. The F50 TCR interacts differently with HLA-A*02:01-M1 than JM22 TCR, and makes less contact with the peptide than JM22 TCR [36]. Altogether, this shows that the CDR3β loop conserved Arg98, has an important contribution to the hydrogen network, and is the best molecular solution to interact with the “vanilla” HLA-A*02:01-M1 complex.

### 4.3. Crushing the Peptide: Binding at All Costs

Most of the tumour-derived CD8+ T cell epitopes for which we have structural information are derived from the NY-ESO-1 (New York esophageal squamous cell carcinoma 1) cancer/testis antigen expressed in a range of cancers, and therefore are a great target for immunotherapeutic design [37,38]. A large amount of work has been done on HLA-A*02:01-restricted epitopes (Table 1), but a recent study focused on the NY-ESO-1 epitope restricted to the HLA-B*07:02 molecule [11]. The study shows that a 13mer peptide, NY-ESO-1-_60–72_ presented by HLA-B*07:02, was able to activate CD8+ T cells (IFNγ production), and in addition described the first structure of a TCR engaging with an HLA-B*07:02 molecule. The structures of two TCRs (KFJ37 and KFJ5) in complex with the HLA-B*07:02-NY-ESO-1-_60–72_ complex showed very different interactions with the pHLA, despite a similar docking angle (70°). Even more striking, the peptide itself was in a completely different conformation when bound to the two TCRs, suggesting that the NY-ESO-1-_60–72_ peptide conformation was flexible in the cleft of HLA-B*07:02. The KFJ37 TCR stabilised the bulged conformation of the peptide, whilst the KFJ5 TCR pushed the peptide down toward the cleft. As a result, the NY-ESO-1-_60–72_ peptide adopted a constrained and helical conformation upon binding of the KFJ5 TCR. This dramatic conformational change of the peptide could explain the fivefold difference in affinity between the KFJ5 (Kd > 200 μM) and KFJ37 (Kd ≈ 40 μM) TCRs.

Altogether, this shows that multiple TCRs can engage with the same epitope, via conserved or mimicked interaction, or by an alternate mode of recognition. In each case, the peptide or the TCR structural plasticity may play a role in structural rearrangement, despite a cost on the overall affinity.

## 5. T Cell Cross-Reactivity: Self-Defence to Self-Sabotage

T cell cross-reactivity is the recognition of at least two unique pMHC complexes by the same TCR, resulting in T cell activation. As there are infinite possibilities of foreign peptides that need to be recognised by a finite number of TCRs within the T cell repertoire, T cells must cross-react to achieve a high level of protection [39]. T cells are like funambulists, walking a fine line between recognising a wide range of foreign peptides while not “falling” and be activated by a self-peptide. The availability of increased TCR–pMHC-I structures brings forth the emergence of mechanisms underpinning T cell cross-reactivity.

### 5.1. Molecular Mimicry Leads to a Million Peptides Recognised and Could Prime Autoimmunity

It is known that when the balance of TCR recognition and reactivity is disrupted, complications such as autoimmunity can arise [40]. The 1E6 T cell clone isolated from a type I diabetic patient recognised the preproinsulin (PPI) signal peptide presented by HLA-A*02:01 [41]. The 1E6 T cell clone was able to recognise 1 million different peptides bound to the HLA-A*02:01 molecule, and the structure of 1E6 TCR in complex with some of those peptides revealed the basis of this multi-ligand recognition [20]. The seven structures with 1E6 TCR complex to HLA-A*02:01 presenting altered PPI ligands showed similarities of recognition mode. This was driven by a conserved ^4^GPD^6^ motif in the peptide interacting with an aromatic cap formed by the TCR CDR3 loops (Tyr97α and Trp97β) (Figure 5A). This illustrates the ability of molecular mimicry to facilitate T cell cross-reactivity through conserved peptide motif. Interestingly, some of the altered PPI peptides were from a pathogenic source, for which the 1E6 TCR possessed higher affinity, such as the RQF peptide (K_d_ of 0.5 μM) from *C. asparagiforme,* than for the PPI peptide (K_d_ >200 μM) [20]. This suggests that pathogen-derived peptides could prime the 1E6 T cell clone and might be responsible for the onset of type I diabetes.

### 5.2. Molecular Mimicry Leads to Off-Target Toxicity

TCRs recognise the composite surface formed by the peptide and MHC molecule. Shared conformation between MHC molecules, and peptides, can lead to two distinct pMHC complexes having a similar surface conformation. As such, two pMHC complexes (for e.g., foreign and self-peptides) can look alike and can no longer be distinguished as being different by TCRs. This results in abnormal TCR recognition due to molecular mimicry. TCRs have a low affinity for foreign peptides (μM range), and are even weaker for self-peptide due to thymic selection. While this is necessary to avoid T cells being activated by healthy cells and causing autoimmunity, in the case of cancer, it does lead to an overall poor T cell response. Typically, TCR affinity towards tumour-associated antigens (TAA) is lower than for pathogenic antigens. One way to improve T cell reactivity towards tumours is to engineer high-affinity TCRs to be used in T cell immunotherapy [9]. However, the fine balance limiting autoimmunity can be disrupted by excessively high TCR affinity. The study on MAGE-A3 epitope (TAA expressed in multiple cancer types) is an example of one such adverse reaction. A TCR recognising the MAGE-A3 peptide was a candidate for the enhanced affinity TCR approach. This modified TCR, called MAG-IC3 TCR, exhibited very high affinity (nM range) for the HLA-A*01:01–MAGE-A3 complex (Table 1). Unfortunately, the MAGE-A3 peptide (EVDPIGHLY) shared five out of nine identical residues with another self-peptide (ESDPIVAQY) derived from the Titin protein on cardiac tissue. Molecular mimicry between the two peptides provoked fatal cardiac toxicity in two patients when treated with the MAG-IC3 enhanced affinity TCR, able to bind the two peptides due to cross-reactivity [42]. The structural comparison of the MAG-IC3 TCR in complex with HLA-A*01:01 presenting either MAGE-A3 or Titin -derived peptide confirmed that MAG-IC3 TCR bound similarly onto both pHLA due to molecular mimicry (Figure 5B). Although the MAG-IC3 TCR had a 10-fold lower affinity for the Titin-derived peptide than the MAGE-A3 peptide (Table 1), the high level of Titin expression in cardiac tissue increased the overall reactivity of the engineered T cell. The lower affinity for Titin might be due to a missing interaction between the TCR and the peptide at P5, where the MAGE-A3 peptide possesses an exposed P5-His, whilst the Titin peptide has a buried P5-Ala. Nevertheless, this illustrates that many factors, and not just affinity, can underlie cross-reactivity, which must be considered when engineering enhanced affinity TCRs.

### 5.3. Cross-Reactivity Becomes Allo-Reactivity: Risk for Transplant Rejection

Prior to organ or tissue transplantation, a close HLA match is sought out; however, a perfect match is near impossible due to the high level of polymorphism of HLAs. After transplantation, the host’s T cells can recognise the donor’s HLA molecule (allo-HLA) as foreign and become activated [43]. Allo-reactivity involves the recognition of both self- and a foreign-peptide in the context of different HLA molecules. There are very few examples where both self- and foreign-derived peptides are known. The most comprehensive example in human is the study of the allo-reactive LC13 TCR [44]. LC13 TCR is specific for HLA-B*08:01 presenting an Epstein–Barr virus (EBV) peptide called FLR [45]. This TCR is shared in HLA-B*08:01/EBV+ individuals (public TCR) and deleted in HLA-B*44:02/03+ individuals due to allo-reactivity with those HLA molecules. The allo-peptide was identified, and structural analysis showed that molecular mimicry was responsible for LC13 TCR allo-reactivity in a peptide-centric manner [44].

Recently, Wang and colleagues showed that T cell allo-reactivity can be specific for both the peptide and MHC molecule simultaneously [46]. The 1406 TCR was isolated from a hepatitis C virus (HCV)-positive/HLA-A*02:01-negative patient who received an HLA-A*02:01-positive liver transplant. The 1406 TCR was shown to be specific for the NS3 derived peptide (KLV) from HCV (Table 1). The 1406 TCR structure in complex with HLA-A*02:01-KLV showed specificity towards the P1-Lys of the peptide (through electrostatic interaction). Interestingly, the structure of the HLA-A*02:01-KLV was similar to the one of HLA-A*02:01-MART-1. The major difference between the two peptides that could impact on 1406 TCR recognition was the critical P1-Lys, which is replaced by a P1-Glu (opposite charge) in the MART-1 peptide. Wang and colleagues mutated the MART-1 at P1 to obtain the MART-1-P1-Lys and tested the ability of 1406 TCR to bind to this modified MART-1 peptide. The SPR analysis revealed that while the 1406 TCR was not able to bind with the MART-1 peptide, it could bind the MART-1-P1-Lys but with lower affinity than the HCV-derived peptide. In this case, both the peptide and HLA molecules were critical for TCR recognition.

Thus far, the molecular basis of TCR allo-reactivity seems to follow the same principles as the one leading to the recognition of viral or pathogenic peptides [47].

There are still aspects of cross-reactivity that we do not understand. T cell cross-reactivity encompasses “the good, the bad, and the ugly” of T cell activation. While it is clearly required that T cells are able to recognise multiple pMHC complexes for an effective immune response, this can also lead to disastrous outcomes. T cell cross-reactivity helps to limit the number of viral escapes due to mutations (the good [12,48]), can underpin autoimmune disease by priming of T cell (the bad [20]), and can even lead to self-destruction such as transplant rejection (the ugly [9]). By understanding such reactivity, we may in the future be able to halt, diminish, or prevent the negative effects of TCR cross-reactivity.

## 6. Beyond αβ TCR: δβ and γδ TCRs Join the Party

It is fairly common to link pMHC-I to αβ TCR recognition; however, recent studies have shown that pMHC-I recognition is not limited to only αβ TCRs. Indeed, we now know how Vδ segments can pair with Jα to form a δα chimeric TCR chain that can be paired with a TCRβ chain. The resulting TCR called either δα/β or δβ TCR can recognise peptides presented by MHC-I as well as lipids bound by MHC-like molecules [17]. In addition, there is an expansion of our knowledge on γδ T cells from recognising empty MHC-like molecule such as T22 [49], to engaging directly with lipid-derived antigens [50,51], and binding pMHC-I complex [19]. It is likely that more information will emerge on the role and antigen repertoire of γδ T cells in the coming years [26,52,53,54].

### 6.1. Chimeric δβ TCR Recognition of pMHC-I Complex

In 2014, the first structure of a chimeric δβ TCR, named clone 12 TCR, in complex with HLA-B*35:01 presenting the CMV derived IPS peptide was solved, alongside a δβ TCR–lipid–CD1d complex [17]. The clone 12 TCR was composed of a TRDV1*01 gene rearranged with the TRAJ52 to form the δα chain and paired with TRBV5-1*01 gene. The particularity of the TRDV1*01 gene is its unusual use of Trp residues in the CDR1δ loop (TS**WW**SYY). Trp is a rare residue in proteins, encoded by a single codon (TGG); is the largest aromatic residue; and is not found in any CDR1 or CDR2 loops in TRAV, TRBV, TRDV, and TRGV other than the TRDV1*01 gene. Interestingly, the four structures of δβ or γδ solved to date in complex with peptide–HLA all share the TRDV1*01 gene (Table 1, [17,18,19]). The clone 12 TCR docks onto HLA-B*35:01-IPS, similarly to a canonical αβ TCR, where the δα chimeric TCR chain interacts with the α2-helix, with the β chain above the α1-helix [17]. Interestingly, the two Trp residues from CDR1δ contact both α1- and α2-helices, dominating the overall interaction for the pHLA complex (27% of TCR BSA; Figure 5C). Subsequently, two more δβ TCR structures have been solved in complex with HIV-derived peptide HLAs (Table 1) [18]. The TU55 TCR is restricted to HLA-B*35:01 while the S19-2 TCR is restricted to HLA-A*24:02. Despite recognising different pHLAs and using different TRBV genes (Table 1), their CDR1δ binds similarly over the pHLA complex, but differently to clone12 TCR (Figure 5C). Therefore, there is some flexibility in how the δ chain can bind and recognise pHLA complexes.

### 6.2. γδ TCR Recognition of Peptide Antigen Presented by MHC

It has been long thought that γδ T cells, in contrast to αβ T cells, can recognise MHC or MHC-like molecules without the requirement to contact or to even enclose an antigen bound in the cleft [55]. This was demonstrated with the early structures of γδ TCRs solved with MHC-like molecule such as T22 [49]. Even more recently, the structure of a γδ TCR in complex with MHC-like molecule MR1 shows that γδ TCRs are not constrained to bind atop the antigen-binding cleft [26], clearly avoiding any bound metabolite-derived antigen. Interestingly, in 2018, Benveniste and colleagues solved the first γδ TCR–peptide–HLA structure [19], and the only one to date. The 5F3 γδ TCR is composed of a TRDV1*01 and TRGV8*01 TCR genes, and therefore shares the TRDV1*01 usage with the previously solved δβ TCRs mentioned above. The surprising feature of this 5F3 γδ TCR recognition of pHLA complex resides in its orientation or docking topology. Indeed, while the δβ TCRs docked similarly to αβ TCRs, whereby the Vα or Vδ domains are over the HLA α2-helix, the 5F3 γδ TCR docked in the opposite orientation or reversed as per discussed in Section 3 (Figure 2B). The 5F3 Vδ domain is docked above the HLA α1-helix, while the Vγ is above the HLA α2-helix (Figure 5D). While there is a requirement for αβ TCRs to engage in a canonical orientation (Figure 2A), most likely due to their CD8/CD4 co-factor binding, there might not be such a need for γδ TCRs. This freedom of docking topology without compromising T cell signalling, in addition to their high diversity, would increase the number of potential antigens that γδ T cells can recognise and engage with.

## 7. TCR Recognition and T Cell Activation

A major question in the field of TCR–pMHC recognition is whether we can determine if a T cell is likely to be activated in response to pMHC recognition [56]. As such, studies investigating T cell activation typically correlate common parameters that underpin TCR–pMHC interaction, such as docking angles [57], co-factor requirement [15,58,59], association rates [60,61], dissociation rates [61,62], half-life or dwell times [60,63], and even mechanistic interactions [64,65,66,67,68]. As these properties are simplified by the concept of binding affinity, TCR–pMHC interactions are commonly quantified using 3D affinity measurements, such as SPR, that can reveal binding affinity and kinetics. Consistently, high TCR–pMHC affinity has been correlated with T cell activation [59,69,70,71,72,73,74]. Exceptions to this have also been discovered, revealing a portion of T cells that are able to recognise and bind to pMHC but cannot induce T cell activation [57,75,76].

In a study by Sibener et al., T cells were able to bind to fluorescently labelled pMHC tetramer with high affinity but were unable to elicit a T cell response (non-agonist) or able to be stimulated (agonist) [76]. From those T cells, the TCRs TCR6 (agonist, K_d_ of 10 µM) and TCR11 (non-agonist, K_d_ of 1 µM) were isolated [76]. Both TCRs were specific for the same pHLA-II complex; however, only TCR6 was able to be activated [76]. Similarly, another agonist/non-agonist TCR pair, specific for a pHLA-I, named HLA-B*35:01-IPL, was also studied, and its structures were determined (Table 1). Both the agonist TCRαβ (TCR589 [76]) and non-agonist TCRγδ (TU55 or TCR55 [18,76]) bind with similar docking angles onto HLA-B*35:01-IPL and had similar binding affinities, albeit slower dissociation rates for the agonist TCR589 [76]. Using these two sets of agonist/non-agonist TCRs [76], the authors revealed that only the agonist TCRs (TCR6 and TCR589) could form catch bonds. The measure of bond lifetime or dissociation lifetime between TCR–pMHC interactions, which are increased (catch bonds) or decreased (slip bonds) when force is applied [65], was the underlying factor for T cell activation [76]. Taken together, this supports that 2D affinity (cell-to-cell) rather than 3D affinity (SPR) is perhaps a better reflection of the TCR–pMHC interface, and a better correlate for T cell activation. As such, T cell activation is driven by cell-to-cell interaction where traditional determinants of TCR affinity (determined by SPR) may be overruled by cellular factors such as TCR expression level, TCR clustering, and pMHC density and stability, which ultimately play a role in avidity.

Many studies over the past several decades have also investigated features of pMHC molecules that correlate to immunogenicity, i.e., the likelihood that the molecular and structural characteristics of antigen presentation may elicit T cell activation. These studies have been shown to correlate pMHC immunogenicity and immunodominance to properties such as pMHC stability [77,78,79,80,81], pMHC dissociation rates [82,83], pMHC density [84], and presence of specific peptide motifs [85].

Although looking at immunogenicity predictors solely from the pMHC perspective can be useful in understanding activation, it only considers one side of the interaction. Structures of pMHC molecules have been traditionally used to provide insight into how antigenic peptides are displayed by MHC. However, pMHCs have been shown to be highly dynamic and may not interact in a way that we can predict. For example, Coles and colleagues structurally characterised 3 TCRs (S1, S2, S3) specific for the heterocyclic peptide variant of NY-ESO-1, SLLMWITQV, presented by the HLA-A*02:01 molecule [86]. The NYE_S1, NYE_S2, and NYE_S3 TCRs bound the pHLA with a *k*_d_ of 7.0, 7.1, and >82 µm, respectively, with fast kinetics [86].

The TCRs bind with a similar docking angle and footprints with a centred pHLA mode of binding. The structures for NYE_S1 and NYE_S2 TCRs show recognition of a bulged peptide peg motif (P4-Met and P5-Trp). Interestingly, the structure for NYE_S3 TCR displayed a novel peptide conformation despite sharing binding to the same centric peptide residues. Upon binding of NYE_S3 TCR, the presentation of centric peptide residues (P5-Trp and P6-Ile) were flipped upon NYE_S3 TCR binding, resulting in P5-Trp to become buried in HLA-A*02:01 cleft and P6-Ile side chain to interact with NYE_S3 TCR. Despite an 11-fold decrease in affinity for NYE_S3 TCR compared to the other TCRs (Table 1), the corresponding T cell was activated (IFNγ release). This shows, again, the difference in TCR affinity does not always differentiate the likelihood of T cell activation. Furthermore, this study highlights the dynamic nature of pMHC molecules and that additional structural conformations can occur during TCR binding. This is in line with an in silico approach study showing that conformational plasticity within TCR–pMHC-I interactions and also peptide–MHC-I interactions initially thought to stabilise the complex were only transient and actually prone to high flexibility [87].

## 8. Conclusions

The last five years have been the most prolific years for the field of TCR–pMHC-I structures and have greatly expanded the current knowledge on TCR recognition. Structural biology has helped uncover TCR diversity in their docking mode, mechanisms of recognition, and affinity ranges, and we expect that this increase will be sustained over the coming years. This will help us to better understand T cells and provide much needed information to help fight known or emerging pathogens such as the new coronavirus responsible for the current COVID-19 pandemic. Despite the tremendous effort by so many groups to advance our understanding of TCR recognition and T cell activation, a large amount still remains unclear.

There is a certain lack of diversity within the current database available. For example, only 20 MHC allomorphs have been solved in complex with a TCR, among them only one HLA-C molecule [14], nothing for HLA-G or HLA-F, and only one γδ TCR [19]. Almost half of the TCR–pMHC-I structures solved involve the HLA-A*02:01 molecule (Table 1, Figure 1D), which is common in Caucasian populations. We hope that future studies will focus on HLAs that have not been extensively studied, which can be applicable for a broader population representing a true cohort of the world’s population, providing meaningful and insightful perspectives into TCR recognition and its medical applications.

The recent structure of CD3–TCR complex helps us understand how this amazing machinery comes together [88]. It will now be required for us to gain an insight into the dynamic changes occurring upon TCR engagement, as well as the requirement for T cell activation, in order to fully appreciate the intricate and specific nature of TCR recognition. Recent examples, such as the novel reverse docking TCRs, provide the tools to deconvolute and uncouple TCR docking and T cell activation [15], which will help clarify key information on T cell selection and activation. While the field has been focused on highly represented TCRs within the immune repertoire, sometimes public or biased, there are certainly a lot of value and information to gain from less represented T cells.

The studies discussed thus far have mostly attempted to isolate specific parameters that may induce T cell receptor signalling from peptide–MHC binding in select systems involving only a handful of TCRs or a specific pMHC molecule. The search for a unifying parameter that underpins T cell activation from all studied pMHC profiles has never been attempted. Interestingly, a study looking at grouping TCRs on the basis of epitope-specific repertoires was able to do so by and clustering or distancing TCRs using a distance-based classifier on the basis of shared motifs and comparing amino acids sequences within CDR loop regions for multiple pMHC systems [89]. Although the study did not directly look at T cell activation, one could expect that only a portion of TCRs within each distinct cluster would lead to T cell activation, whilst the rest may be inhibited by poor interactions. As such, this tool provides a means to be able to investigate distinct TCR repertoires towards multiple pMHC systems.

**Table 1 ijms-22-00068-t001:** All unique TCR–pMHC-I structures available.

MHC	Peptide	TCR	TRAV	TRBV	PDB	CDR3α	CDR3β	BSA (Å^2^)	Vα (%)	Vβ (%)	Pep (%)	Kd (μM)	Ref
HLA-A*02:01	LLFGYPVYV (HTLV)	A6	12-2*02	6-5*01	1AO7	AVTTDSWGKLQ	ASRPGLAGGRPEQY	1990	65	35	34	0.9	[8]
HLA-A*02:01	LLFGYPVYV (HTLV)	B7	29/DV5*01	6-5*01	1BD2	AAMEGAQKLV	ASSYPGGGFYEQY	1710	68	32	32	ND	[90]
H2-Kb	EQYKFYSV (self)	2C	9-4*01	13-2*01	2CKB	AVSGFASALT	ASGGGGTLY	2100	56	44	25	100	[91]
H2-Kb	INFDFNTI (self)	BM3.3	16/DV11*01	1*01	1FO0	AMRGDYGGSGNKLI	TCSADRVGNTLY	1240	37	63	19	2.6	[92]
H2-Kb	KVITFIDL (self)	KB5-C20	14-1*01	1*01	1KJ2	AARYQGGRALI	TCSAAPDWGASAETLY	1890	48	52	21	ND	[93]
HLA-A*02:01	ALWGFFPVL (self)	AHIII 12.2	12D-2*01	13-3*01	1LP9	ALFLASSSFSKLV	ASSDWVSYEQY	1840	56	44	26	11.3	[94]
HLA-B*08:01	FLRGRAYGL (EBV)	LC13	26-2*01	7-8*01	1MI5	ILPLAGGTSYGKLT	ASSLGQAYEQY	2020	57	43	18	10	[45]
HLA-A*02:01	GILGFVFTL (Influenza)	JM22	27*01	19-01	1OGA	AGAGSQGNLI	ASSSRSSYEQY	1470	33	67	28	5.6	[34]
HLA-B*35:08	LPEPLPQGQLTAY (EBV)	SB27	19*01	6-1*01	2AK4	ALSGFYNTDKLI	ASPGLAGEYEQY	1750	58	42	43	9.9	[95]
HLA-A*02:01	SLLMWITQC (self)	1G4	21*01	6-5*01	2BNR	AVRPTSGGSYIPT	ASSYVGNTGELF	1920	48	52	38	13.3	[96]
HLA-E*01:03	VMAPRTLIL (CMV)	KK50.4	26-1*01	14*01	2ESV	IVVRSSNTGKLI	ASSQDRDTQY	1810	39	61	27	30.2	[97]
HLA-B*35:01	EPLPQGQLTAY (EBV)	ELS4	1-2*01	10-3*01	2NX5	AVQASGGSYIPT	ATGTGDSNQPQH	2400	43	57	25	30	[98]
H2-Ld2	QLSPFPFDL (synthetic)	2C	9-4*01	13-2*01	2OI9	AVSGFASALT	ASGGGGTLY	1710	52	48	31	2	[99]
H2-Kbm8	SQYYYNSL (synthetic)	BM3.3	16/DV11*01	1*01	2OL3	AMRGDYGGSGNKLI	TCSADRVGNTLY	1710	52	48	30	112	[100]
HLA-B*44:05	EENLLDFVRF (EBV)	DM	26-1*02	7-9*01	3DXA	IVWGGYQKVT	ASRYRDDSYNEQF	2090	52	48	33	0.3	[101]
HLA-B*08:01	FLRGRAYGL (EBV)	CF34	14/DV4*01	11-2*01	3FFC	AMREDTGNFY	ASSFTWTSGGATDTQY	2170	42	58	20	8.9	[102]
HLA-A*02:01	NLVPMVATV (CMV)	RA14	24*01	6-5*01	3GSN	ARNTGNQFY	ASSPVTGGIYGYT	1930	52	48	33	27	[32]
HLA-A*02:01	ELAGIGILTV (MART-1)	CD8	12-2*01	30*01	3HG1	AVNVAGKST	AWSETGLGTGELF	2030	48	52	26	18	[103]
HLA-B*44:05	EEYLQAFTY (self)	LC13	26-2*01	7-8*01	3KPS	ILPLAGGTSYGKLT	ASSLGQAYEQY	2330	53	47	22	49	[44]
HLA-B*35:01	HPVGEADYFEY (EBV)	TK3	20*01	9*01	3MV7	AVQDLGTSGSRLT	ASSARSGELF	2040	55	45	42	2.2	[104]
HLA-A*02:01	GLCTLVAML (EBV)	AS01	5*01	20-1*01	3O4L	AEDNNARLM	SARDGTGNGYT	2150	54	46	25	8.1	[105]
H2-Db	SSLENFRAYV (Influenza)	6218	21/DV12*02	29*01	3PQY	ILSGGSNYKLT	ASSFGREQY	1696	50	50	39	2	[106]
HLA-A*02:01	AAGIGILTV (MART-1)	DMF5	12-2*01	6-4*01	3QDJ	AVNFGGGKLI	ASSLSFGTEAF	2240	59	41	23	40	[107]
HLA-A*02:01	LAGIGILTV (MART-1)	DMF4	35*01	10-3*01	3QDM	AGGTGNQFY	AISEVGVGQPQH	1750	48	52	33	170	[107]
H2-Kb	WIYVYRPM (synthetic)	YAe62	6D-3*01	13-2*01	3RGV	AANSGTYQR	ASGDFWGDTLY	1650	44	56	23	15	[108]
HLA-B*57:03	KAFSPEVIPMF (HIV)	AGA1	5*01	19*01	2YPL	AVSGGYQKVT	ASTGSYGYT	1580	69	31	49	3	[109]
HLA-B*08:01	FLRGRAYGL (EBV)	RL42	12-1*01	6-2*01	3SJV	VVRAGKLI	ASGQGNFDIQY	2110	64	36	20	31	[110]
HLA-A*02:01	ALWGPDPAAA (PPI)	1E6	12-3*01	12-4*01	3UTS	AMRGDSSYKLI	ASSLWEKLAKNIQY	1380	49	51	34	278	[41]
HLA-A*24:02	RFPLTFGWCF (HIV)	C1-28	8-3*01	4-1*01	3VXM	AVGAPSGAGSYQLT	ASSPTSGIYEQY	1970	78	22	38	21.1	[111]
HLA-A*24:02	RYPLTFGWCF (HIV)	H27-14	21*01	7-9*01	3VXR	AVRMDSSYKLI	ASSSWDTGELF	1930	46	54	37	9.7	[111]
HLA-A*24:02	RFPLTFGWCF (HIV)	T36-5	12-2*01	27*01	3VXU	WGTYNQGGKLI	ASSGASHEQY	2390	45	55	27	1.6	[111]
HLA-B*27:05	KRWIILGLNK (HIV)	C12C	14/DV4*02	6-5*01	4G8G	AMRDLRDNFNKFY	ASREGLGGTEAF	1860	58	42	35	2	[112]
HLA-B*35:08	LPEPLPQGQLTAY (EBV)	SB47	39*01	5-6*01	4JRY	AVGGGSNYQLI	ASSRTGSTYEQY	2010	44	56	17	25	[113]
HLA-B*51:01	TAFTIPSI (HIV)	3B	17*01	7-3*01	4MJI	ATDDDSARQLT	ASSLTGGGELF	2230	55	45	13	81.8	[114]
HLA-B*35:01	IPSINVHHY (CMV)	Clone12	TRDV1*01	5-1*01	4QRR	ALGELAGAGGTSYGKLT	ASSLEGGYYNEQF	2040	65	35	25	15	[17]
HLA-B*08:01	HSKKKCDEL (HCV)	DD31	9-2*01	11-2*01	4QRP	ALSDPVNDMR	ASSLRGRGDQPQH	2351	53	47	23	2.5	[115]
H2-Ld2	GAPWNPAMMI (p3M11I)	M33	9-4*01	13-2*01	4NHU	AVSLHRPALT	ASGGGGTLY	1504	71	29	18	ND	[NA]
HLA-A*02:01	NLVPMVATV (CMV)	C7	24*01	7-2*02	5D2L	AFITGNQFY	ASSQTQLWETQY	1901	56	44	33	5.1	[33]
HLA-A*02:01	NLVPMVATV (CMV)	C25	26-2*01	7-6*01	5D2N	ILDNNNDMR	ASSLAPGTTNEKLF	1832	36	64	30	4.7	[33]
HLA-A*01:01	EVDPIGHLY (MAGE-A3)	MAG-IC3	21*01	5-1*01	5BRZ	AVRPGGAGPFFVV	ASSFNMATGQY	1817	62	38	22	0.0071	[9]
HLA-A*01:01	ESDPIVAQY (Titin)	MAG-IC3	21*01	5-1*01	5BS0	AVRPGGAGPFFVV	ASSFNMATGQY	2048	58	42	26	0.0767	[9]
HLA-A*02:01	YQFGPDFPIA (synthetic)	1E6	12-3*01	12-4*01	5C07	AMRGDSSYKLI	ASSLWEKLAKNIQY	1697	41	59	37	7.4	[20]
HLA-A*02:01	RQWGPDPAAV (synthetic)	1E6	12-3*01	12-4*01	5C08	AMRGDSSYKLI	ASSLWEKLAKNIQY	1590	45	55	34	7.8	[20]
HLA-A*02:01	MVWGPDPLYV (*B. fragilis*)	1E6	12-3*01	12-4*01	5C0A	AMRGDSSYKLI	ASSLWEKLAKNIQY	1364	37	63	43	600	[20]
HLA-A*02:01	RQFGPDFPTI (*C. asparagiforme*)	1E6	12-3*01	12-4*01	5C0B	AMRGDSSYKLI	ASSLWEKLAKNIQY	1672	43	57	33	0.5	[20]
HLA-A*02:01	RQFGPDWIVA (synthetic)	1E6	12-3*01	12-4*01	5C0C	AMRGDSSYKLI	ASSLWEKLAKNIQY	1681	43	57	37	44.4	[20]
H2-Db	ASNENMETM (Influenza)	NP2-B17	14-1*01	17*01	5SWS	AASEGSGSWQLI	ASSAGLDAEQY	1900	28	72	17	30	[15]
HLA-E*01:03	VMAPRTLIL (CMV)	GF4	35*02	9*01	5W1V	AGQPLGGSNYKLT	ASSANPGDSSNEKLF	2129	49	51	21	37.4	[116]
HLA-A*11:01	GTSGSPIVNR (Dengue)	D30	30*01	11-2*01	5WKF	GLGDAGNMLT	ASSLGQGLLYGYT	1769	31	69	25	136	[10]
H2-Db	SQLLNAKYL (Malaria)	NB1	8-2*01	13-3*01	5WLG	ATVYAQGLT	ASSDWGDTGQLY	2168	61	39	24	19	[29]
HLA-B*35:01	IPLTEEAEL (HIV)	TU55	TRDV1*01	6-1*01	5XOT	ALGEGGAQKLV	ASRTRGGTLIEQY	1964	50	50	28	0.27	[18]
HLA-A*24:02	RYPLTFGWCF (HIV)	S19-2	TRDV1*01	30*01	5XOV	ALGELARSGGYQKVT	AWSVSVGAGVPTIY	1818	75	25	34	1.6	[18]
HLA-A*02:01	GILGFVFTL (Influenza)	LS01	24*01	19*01	5ISZ	AFDTNAGKST	ASSIFGQREQY	1842	34	66	29	30	[35]
HLA-A*02:01	GILGFVFTL (Influenza)	LS10	38-2/DV8*01	19*01	5JHD	AWGVNAGGTSYGKLT	ASSIGVYGYT	1972	47	53	24	32	[35]
HLA-A*02:01	KLVALGINAV (HCV)	1406	38-2/DV8*01	25-1*01	5JZI	AYGEDDKII	ASRRGPYEQY	2082	55	45	32	16	[46]
HLA-A*02:01	ILAKFLHWL (self)	ILA1	22*01	6-5*01	5MEN	AVDSATSGTYKYI	ASSYQGTEAF	2276	61	39	33	48	[117]
HLA-A*02:01	SLYNTVATL (HIV)	868	12-2*01	5-6*01	5NME	AVRTNSGYALN	ASSDTVSYEQY	2118	56	44	24	0.082	[118]
HLA-A*02:01	GILGFVFTL (Influenza)	F50	13-1*02	27*01	5TEZ	AASFIIQGAQKLV	ASSLLGGWSEAF	1745	47	53	25	76	[36]
H2-Db	KAVYNFATM (LCMV)	P14	14D-1*01	13-3*01	5TJE	AALYGNEKIT	ASSDAGGRNTLY	1582	49	51	19	ND	[119]
HLA-A*02:01	EAAGIGILTV (MART-1)	199.54-16	12-2*02	19*01	5NHT	AVGGGADGLT	ASSQGLAGAGELF	1792	51	49	30	ND	[NA]
HLA-A*02:01	KLVALGINAV (HCV)	TCR	38-1*01	25-1*01	5YXN	AYGEDDKII	ASRRGPYEQY	2195	56	44	31	ND	[NA]
HLA-A*02:01	SMLGIGIVPV (synthetic)	DMF5	12-2*01	6-4*01	6AM5	AVNFGGGKLI	ASSLSFGTEAF	2049	57	43	25	43	[16]
HLA-A*02:01	MMWDRGLGMM (synthetic)	DMF5	12-2*01	6-4*01	6AMU	AVNFGGGKLI	ASSLSFGTEAF	1719	55	45	31	32	[16]
HLA-B*07:02	APRGPHGGAASGL (NY-ESO-1)	KFJ5	4*01	28*01	6AVF	LVGEILDNFNK	ASSQRQEGDTQY	1574	75	25	39	>200	[11]
HLA-B*07:02	APRGPHGGAASGL (NY-ESO-1)	KFJ37	4*01	9*01	6AVG	LVVDQKLV	ASSGGHTGSNEQF	2341	44	56	36	41	[11]
HLA-B*35:01	IPLTEEAEL (HIV)	TCR589	19*01	5-4*01	6BJ2	ALSHNSGGSNYKLT	ASSFRGGKTQY	2110	58	42	28	4	[76]
HLA-A*02:01	ELAGIGILTV (MART-1)	5F3	TRDV1*01	TRGV8*01	6D7G	ATWASSDWIKT	ALGELGWDTDKLI	1888	57	43	16	2.9	[19]
HLA-B*37:01	FEDLRVLSF (Influenza)	EM2	30*01	19*01	6MTM	GTERSGGYQKVT	ASSMSAMGTEAF	1786	40	60	23	137	[12]
H2-Db	KAPYDYAPI (self)	P14	14D-1*01	13-3*01	6G9Q	AALYGNEKIT	ASSDAGGRNTLY	1602	52	48	31	ND	[NA]
HLA-A*02:06	KQWLVWLFL (TIL)	302TIL	38-2/DV8*01	12-3*01	6P64	AFMDSNYQLI	ASSRTSPTDTQYF	1658	69	31	43	ND	[13]
HLA-A*02:01	SLSKILDTV (NY-BR-1)	NYBR	22*01	11-2*01	6R2L	AVGGNDWNTDKLIF	ASSPLDVSISSYNEQFF	2569	63	37	25	ND	[NA]
HLA-A*02:01	SLLMWITQV (NY-ESO-1)	NYE_S1	12-2*01	6-5*01	6RPB	AVKSGGSYIPT	ASSYLNRDSALD	1841	28	72	36	7	[86]
HLA-A*02:01	RMFPNAPYL (WT1)	a7b2	26-1*01	7-9*03	6RSY	IGGGTTSGTYKYIF	ASSLGFGRDVMRF	2206	45	55	26	0.07	[120]
HLA-C*08:02	GADGVGKSAL (TIL)	TCR10	12-02*01	10-2*01	6UON	AAAMDSSYKLIF	ASSDPGTEAFF	1529	67	33	20	6.7	[14]
HLA-A*02:01	HMTEVVRHC (p53)	38-10	38-1*01	10-3*01	6VRN	AFMGYSGAGSYQLTF	AISELVTGDSPLHF	1736	58	42	40	39.9	[121]
HLA-B*07:02	RPPIFIRRL (EBV)	HD14	24*01	4-1*01	6VMX	AFGSSNTGKLI	ASSQDLFTGGYT	1756	55	45	35	1.21	[122]
HLA-A*02:01	SLLMWITQV (NY-ESO-1)	NYE_S3	12-2*01	7-6*01	6RP9	ALTRGPGNQFY	ASSSPGGVSTEAF	1335	31	69	44	>82	[86]
HLA-A*02:01	SLLMWITQV (NY-ESO-1)	NYE_S2	3*01	29-1*01	6RPA	AVRDINSGAGSYQLT	SVGGSGGADTQY	2230	42	58	30	7.1	[86]
HLA-A*02:01	GVYDGREHTV (MAGE-4)	GVY01	10*01	28*01	6TRO	VVNHSGGSYIPTF	ASSFLMSGDPYEQYF	1798	68	32	25	0.19	[123]
HLA-A*02:01	HMTEVVRHC (p53)	1a2	12-3*01	27*01	6VQO	AMSGLKEDSSYKLIF	CASSIQQGADTQYF	1462	72	28	43	16.2	[121]
HLA-A*02:01	HMTEVVRHC (p53)	12-6	12-1*01	6-1*01	6VRM	VVQPGGYQKVTF	ASSEGLWQVGDEQYF	1588	50	50	37	1.1	[121]
							**Average**	**1885**	**52**	**48**	**29**	**40**	

TIL: tumour-infiltrating T lymphocytes; WT1: Wilms tumour protein; TRAV: TCR Variable (V) α chain; TRBV: TCR Variable (V) β chain; TRDV: TCR Variable δ chain; TRGV: TCR Variable γ chain; BSA: buried surface area; Pep: peptide; NA: not available; ND: not determined. The Protein Data Bank (PDB) structures highlighted in yellow were previously analysed [124], and those highlighted in blue are the newly analysed structures in this review.

Despite efforts to compare the biochemical and structural parameters of all existing TCR–pMHC interactions, we are still at a loss for determining a unified correlate that leads to effective T cell activation. Interestingly, our analysis of the parameters listed in Table 1 have revealed that there is a correlation between the peptide contribution and BSA, as well as between TCR affinity and BSA (Figure 6). The correlation between the peptide contribution and BSA is inversely proportional. As the peptide presented by MHC-I molecule is limited in length due to the closed end of the MHC-I cleft (Figure 1A), the overall peptide surface is also limited by the volume of the MHC antigen-binding cleft. Therefore, if the overall contact surface at the TCR and pMHC-I interface increases, it is mainly due to an increase of MHC-I BSA, which will in turn decrease the peptide contribution (Figure 6B). We found a significant positive correlation between the TCR affinity and the TCR–pMHC-I BSA, although one could expect that an increase in the number of molecular interactions and contacts, no matter their strength, could potentially contribute and accumulate towards a higher affinity (Figure 6D). As affinity is an important parameter for T cell activation, but not the sole driver, we here present a direct link between the structural parameters of the TCR–pMHC-I complexes that impact T cell function. Future studies would benefit from an integration of the structural and functional parameters to uncover the link between TCR recognition and T cell activation.

## Figures and Tables

**Figure 1 ijms-22-00068-f001:**
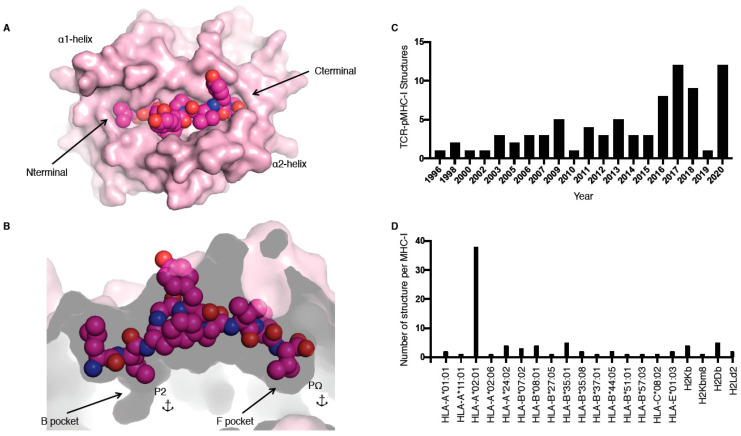
pMHC-I structure and current T cell receptor (TCR)-pMHC-I structures available. (**A**,**B**) Cleft of major histocompatibility complex (MHC)-I molecule (pale pink) represented as surface from a top-down view (**A**) and side view (**B**); the peptide is represented as pink spheres with the anchors residues at position 2 (P2) and at the last position (PΩ) in the B and F pockets (**B**). (**C**,**D**) Number of TCR–pMHC-I structures solved per year (**C**) or per MHC-I (**D**).

**Figure 2 ijms-22-00068-f002:**
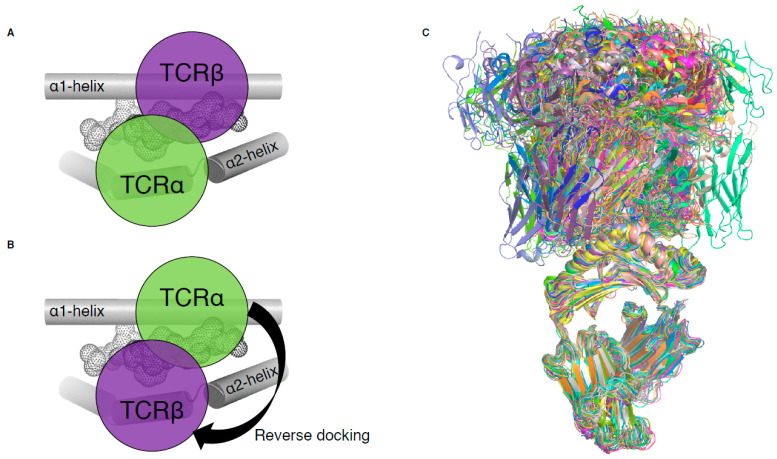
TCR docking on pMHC-I complex. (**A**) Schematic of canonical TCR docking on pMHC-I, whereby the TCRα (green) is above the MHC-I α2-helix (grey tube) and the TCRβ chain (purple) is above the MHC-I α1-helix (grey tube). (**B**) Schematic of reversed docking for TCR, with the arrow indicating a 180° shift from the canonical docking mode (shown on (**A**)). Here, the TCRα (green) is above the α1-helix, while the TCRβ chain (purple) is above the α2-helix. The peptide is represented as dots on panels (**A**,**B**). (**C**) Overlay of all TCR–pMHC-I structures from Table 1, aligned on the MHC-I antigen binding cleft (residues 1–180).

**Figure 3 ijms-22-00068-f003:**
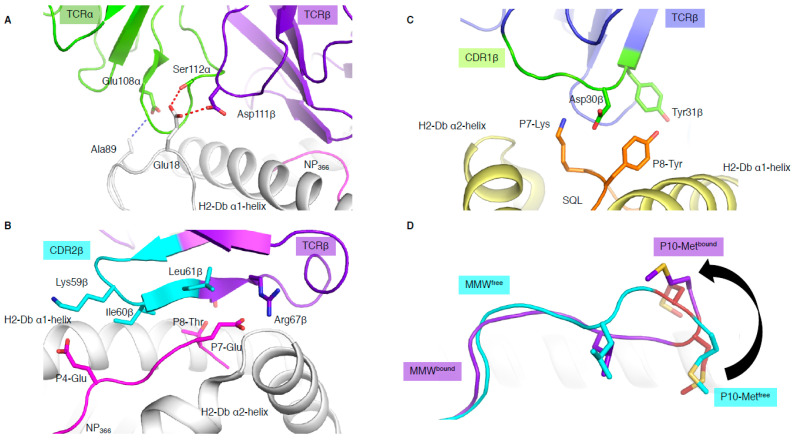
Unconventional TCR docking or TCR–peptide interactions**.** (**A**) NP2-B17 TCR (α in green and β in purple) contacting the loops outside the H-2D^b^ antigen-binding cleft, and, namely, the residues 18 and 89 (white stick). The red and blue dashed lines represent the hydrogen and Van der Waals interactions. (**B**) The NP2-B17 TCR CDR2β (cyan) and framework β (purple) interact with the NP_366_ peptide (pink) presented by H-2D^b^ (white). (**C**) TRBV13-3+ TCR (β chain in blue) with the CDR1β (green) residues (sticks) interacting specifically with the malaria-derived SQL peptide (orange) P7-Lys and P8-Tyr (sticks) presented by the H-2D^b^ (yellow). (**D**) Comparison of the MMW peptide structure (loop) presented by human leukocyte antigen (HLA)-A*02:01 without (cyan) or with the DMF5 TCR (purple). The DMF5 TCR binding leads to a register shift of the MMW peptide, whereby the P10-Met^free^ in the HLA cleft (cyan sticks) is flipped out of the cleft (P10-Met^bou^nd in purple stick) upon binding of the DMF5 TCR.

**Figure 4 ijms-22-00068-f004:**
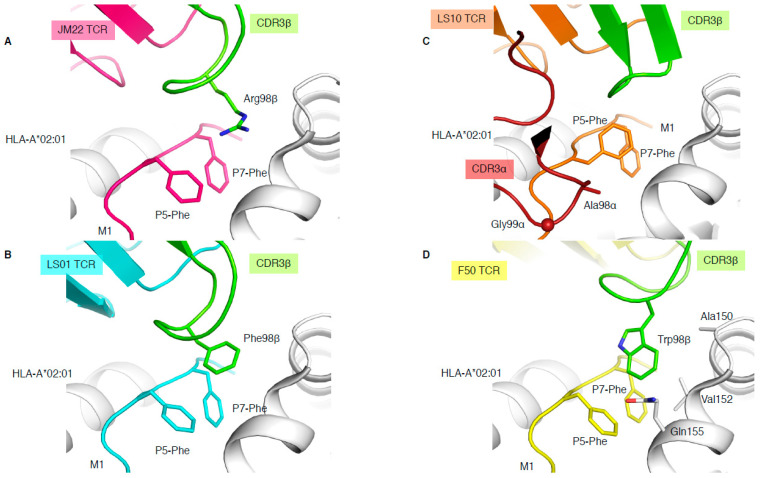
Different solutions to engage with the same pMHC-I complex**.** Structures of HLA-A*02:01 (white) presenting the M1 peptide in complex with the JM22 TCR (**A**), LS01 TCR (**B**), LS10 TCR (**C**), or F50 TCR (**D**). The structures were superimposed by aligning the cleft (residues 1–180) and presented in the same orientation. The M1 peptide is coloured as per the bound TCR in pink with JM22 (**A**), cyan with LS01 (**B**), orange with LS10 (**C**), and yellow with F50 (**D**), and the P5-Phe and P7-Phe are represented as sticks. The CDR3β loop is coloured in green for all TCRs, and the CDR3α is removed from all panels but (**C**), where its coloured in red. The sphere on panel (**C**) represents the Cα atom of the F50 TCR Gly99α.

**Figure 5 ijms-22-00068-f005:**
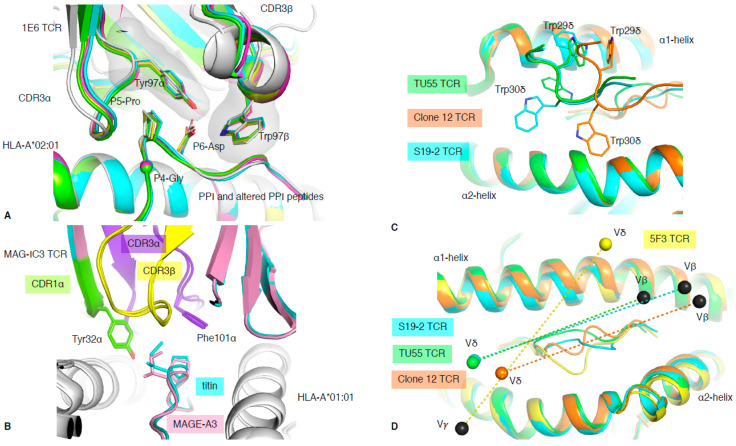
TCR cross-reactivity and non-αβ TCR recognition. (**A**) Structural overlay of the 1E6 TCR in complex with the PPI (white) and altered PPI peptides: YQF (green), RQW (orange), MVW (cyan), RQF-I (pink), and RQF-A (yellow). The conserved ^4^GPD^6^ motif in the peptide is represented as sticks and spheres for the Cα atom of the P4-Gly. The CDR3 loop residue Tyr97α and Trp97β forming the hydrophobic cap are represented as sticks. (**B**) Structural overlay of the MAG-IC3 TCR in complex with the HLA-A*01:01 (white) presenting the MAGE-A3 (pink) or titin (cyan) peptide. The TCR is coloured as per the bound peptide, with the CDR1α in green, CDR3α in purple, and CDR3β in yellow. (**C**) Top view of structural overlay of δβ TCRs in complex with pHLA-I complex. The TU55 TCR-HLA-B*35:01-IPL is in green, S19-2 TCR-HLA-A24:02-RYP is in cyan, and clone 12 TCR-HLA-B*35:01-IPS is in orange. The conserved TRDV1*01 CDR1δ is represented as a loop with the Trp29δ and Trp30δ represented as sticks. (**D**) As per panel (**C**), the top view of the MHC-I cleft with a structural overlay of three δβ TCRs–pMHC-I complex and the 5F3 γδ TCR added (yellow). The mass centres for the Vβ or Vγ are represented as black spheres, while the Vδ mass centres are presented as coloured spheres matching each TCR as per panels (**C**,**D**). All the structural overlays are aligned on the MHC-I cleft (residues 1–180).

**Figure 6 ijms-22-00068-f006:**
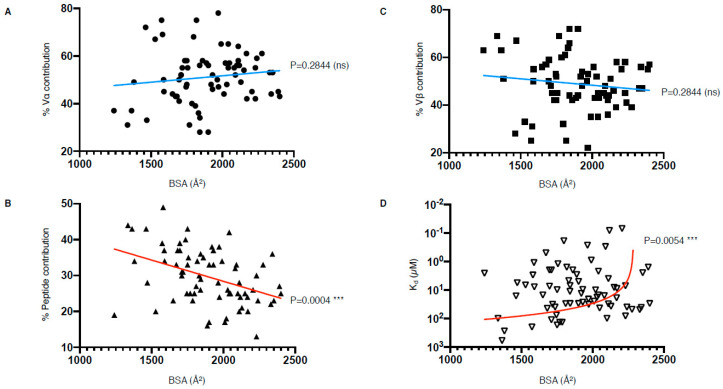
TCR–pMHC-I structural and biophysical parameter correlation. (**A**–**D**) Correlation between buried surface area (BSA); contribution from the Vα, Vβ, or peptide; as well as affinity (Kd) were assessed using simple linear regression calculated with Prism 8 (*p* ≤ 0.05 is considered as significant, ns: *p* > 0.05 ***: *p* ≤ 0.001). Derived function is coloured red if the correlation is significant and blue if not significant. The TCR–pMHC-I complexes without reported affinity (NA in Table 1) were removed as well as the engineered high affinity MAG-IC3 TCR to keep only naturally occurring TCR.

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
