# Peer review of "TCR Recognition of Peptide–MHC-I: Rule Makers and Breakers"

_ijms, 2020, doi:10.3390/ijms22010068_

Round 1

Reviewer 1 Report

In this well written and comprehensive review of TCR recognition of pMHC class I ligands, Gras and colleagues focus on TCR-pMHC class I structures determined in the last 5 years. As the authors point out, more than half of all TCR-pMHC class I complexes were solved during this period, which justifies a review. They emphasize new insights arising from these structures, such as reversed polarity TCRs, TCR cross-reactivity, and pMHC recognition by gamma/delta TCRs.

Points to address:

  1. On p. 2, line 64, “Bjorkman and colleagues” should be corrected to “Wiley and colleagues.” Bjorkman was not an author on the first reported TCR-pMHC class I structure (ref. 8).
  2. On p. 12, 1st paragraph, the authors imply that MAGE-A3 is a neoantigen. This is incorrect. MAGE-A3 is a non-mutated tumor-associated self antigen.
  3. Regarding cancer neoantigens, the authors should add a section summarizing very recent structural studies of TCR recognition of this type of neoepitope:

High-affinity oligoclonal TCRs define effective adoptive T cell therapy targeting mutant KRAS-G12D. Sim MJW, Lu J, Spencer M, Hopkins F, Tran E, Rosenberg SA, Long EO, Sun PD. Proc Natl Acad Sci USA. 2020 Jun 9;117(23):12826-12835. doi: 10.1073/pnas.1921964117.

Structural basis for oligoclonal T cell recognition of a shared p53 cancer neoantigen. Wu D, Gallagher DT, Gowthaman R, Pierce BG, Mariuzza RA. Nat Commun. 2020 Jun 9;11(1):2908. doi:10.1038/s41467-020-16755-y.

Structural dissimilarity from self drives neoepitope escape from immune tolerance. Devlin JR, Alonso JA, Ayres CM, Keller GLJ, Bobisse S, Vander Kooi CW, Coukos G, Gfeller D, Harari A, Baker BM. Nat Chem Biol. 2020 Nov;16(11):1269-1276. doi: 10.1038/s41589-020-0610-1.

Cancer neoantigens are currently under intense investigation for adoptive T cell therapy and vaccination, and so would be of interest to many readers of this review.

4. On p. 12, lines 361-362, the authors state: “Typically, TCR affinity towards neoantigens (cancer derived antigenic peptides) is lower than for pathogenic antigens.” However, TCR affinities for the bona fide cancer neoantigens described in the above three references are in the micromolar range (i.e. comparable to the affinities for microbial antigens). TCRs typically have low affinity for non-mutated self antigens, such as MAGE-A3.

5. My version of the manuscript has only 89 references. However, Table 1 includes references up to 123. Where are the missing references?

Author Response

We would like to thanks the reviewers for their carefully reading of our reviews, their positive comments, and criticism that we believe have strengthen our manuscript.

REVIEWER #1

  1. On p. 2, line 64, “Bjorkman and colleagues” should be corrected to “Wiley and colleagues.” Bjorkman was not an author on the first reported TCR-pMHC class I structure (ref. 8).

This has been changed, thanks for pointing this out, it now reads

The first TCR-pMHC-I structure solved was by Garboczi and colleagues using X-ray crystallography in 1996 for HLA-A*02:01 presenting the Tax peptide from human T cell lymphotropic virus in complex with the A6 TCR (Table 1) 8.

  1. On p. 12, 1st paragraph, the authors imply that MAGE-A3 is a neoantigen. This is incorrect. MAGE-A3 is a non-mutated tumor-associated self antigen.

This has been changed, and now reads:

Typically, TCR affinity towards tumour associated antigens (TAA) is lower than for pathogenic antigens.

  1. Regarding cancer neoantigens, the authors should add a section summarizing very recent structural studies of TCR recognition of this type of neoepitope …  Cancer neoantigens are currently under intense investigation for adoptive T cell therapy and vaccination, and so would be of interest to many readers of this review.

We agree that cancer neo-antigens is a hot and exciting topic in cancer immunotherapy, however it does not align with the scope of our review, as very limited structural information is available on this topic. So far not many structure of TCR-pMHC complexes are relevant to neo-antigens, therefore  we briefly touched on cancer antigens, for e.g. NY-ESO-1 (pg 10) and MAGE-A3 (pg 12), albeit in the context of TCR/pMHC structural plasticity and molecular mimicry, respectively.

  1. On p. 12, lines 361-362, the authors state: “Typically, TCR affinity towards neoantigens (cancer derived antigenic peptides) is lower than for pathogenic antigens.” However, TCR affinities for the bona fide cancer neoantigens described in the above three references are in the micromolar range (i.e. comparable to the affinities for microbial antigens). TCRs typically have low affinity for non-mutated self antigens, such as MAGE-A3.

We agree that leads to confusion, and we have changed to:

Typically, TCR affinity towards tumour associated antigens (TAA) is lower than for pathogenic antigens.

  1. My version of the manuscript has only 89 references. However, Table 1 includes references up to 123. Where are the missing references?

The references have been updated.

Reviewer 2 Report

This review provides an update on the current knowledge of the T-cell receptor structures, their interaction with antigenic peptides in complex with MHC molecules, and the impact on T cell activation. The authors discuss in an exhaustive and critical manner the unique properties of the TCR-MHC/peptide interactions that have been characterized so far. The review is comprehensive, including the description of structures identified/characterized in the past three years. Globally, the review is well written, however there are several points that need a particular attention.

Major comments

#1 The abstract section should be revised to cover the full scope of TCR function in adaptive immunity, beyond “recognition of what so called “pathogenic peptides” and “infection”. TCR are not limited to recognition of “pathogenic-derived peptides” but also recognize a wide range of self-peptides (which can lead to autoimmunity) or cancer-associated peptides (=self-antigens) or cancer-specific peptides (=neoantigens and viral antigens i.e. EBV, HCMV, HPV etc…).

#2 In the chapter 1, the are some confusions regarding the terms “pathogens” (lane 28) and “pathogenic epitopes” (lane 30). Pathogenic peptides are derived from pathogens (virus, bacteria, fungi) and in some cases from malignant transformation of healthy cells by viruses (i.e. HPV). Peptides derived from cancer cells are mainly including either self-antigens, (i) non-mutated peptides or (ii) mutated peptides or neoantigens.  Thus, T cells can recognized and respond to a wide range of MHC-peptide complexes.    

Minor comments

#1 The following words or expressions should be avoided.

  • “of course” (lane 39-40)
  • “now” (lane 71)

#2 Figures and table 1 are not publication quality.

#3 Lane 53 - (D) and (J) definitions are not correct. (D) and (J) stand for Diversity and Joining, respectively.

#4 Please explain what the letter omega (Ω) mean in Figure 1 and in the corresponding legend and text? Also precise the abbreviation P for “position” when referring to peptides.

#5 Mistakes to be corrected in the legend of table 1 (page 4): “chain” instead of “chian”. Also the reference 124 does not appear in the reference list?

#6 Fig.3D - the legend should be changed to reflect the abbreviations shown in the figure: In the legend, P10-Met and P9-Met are mentioned while in the figure one could notice a different abbreviation P10-Metfree and P10-Metbound.

#7 Explain why amino-acid abbreviations differ throughout the text. The authors use “one letter symbol” (30DY31, 98RS99, 4GPD6) or “three-letter” abbreviation (Asp30b, Tyr31b, P4-Pro, P5-Met, P8-Thr etc….). Consistency would be helpful.

#8 Structured references would be appreciated, in particular for ref.2: the names & surnames of authors are not abbreviated the same way as the other references. Some authors are named without abbreviations (James McCluskey, Mandvi Bharadwaj, etc…), other only with abbreviations (SG, JR). Sometimes “coma” between author names are missing or doubled. Also a ref. including more than 6 authors should include the abbreviation et al. Finally, the year of publication is not 2009 but 2010.

Author Response

We would like to thanks the reviewers for their carefully reading of our reviews, their positive comments, and criticism that we believe have strengthen our manuscript.

REVIEWER #2

Major comments

#1 The abstract section should be revised to cover the full scope of TCR function in adaptive immunity, beyond “recognition of what so called “pathogenic peptides” and “infection”. TCR are not limited to recognition of “pathogenic-derived peptides” but also recognize a wide range of self-peptides (which can lead to autoimmunity) or cancer-associated peptides (=self-antigens) or cancer-specific peptides (=neoantigens and viral antigens i.e. EBV, HCMV, HPV etc…).

We have modified the abstract to include all peptides that T cell might recognised:

Upon recognition of protein fragments (peptides), activated T cells will contribute to the immune response and help clear infection.

#2 In the chapter 1, the are some confusions regarding the terms “pathogens” (lane 28) and “pathogenic epitopes” (lane 30). Pathogenic peptides are derived from pathogens (virus, bacteria, fungi) and in some cases from malignant transformation of healthy cells by viruses (i.e. HPV). Peptides derived from cancer cells are mainly including either self-antigens, (i) non-mutated peptides or (ii) mutated peptides or neoantigens.  Thus, T cells can recognized and respond to a wide range of MHC-peptide complexes.    

We have modified the start of section 1, page 1:

The peptide presented by MHC can be derived from the host proteins (self-peptides), pathogens (virus and bacteria), or tumours. The T cells need to differentiate between self and foreign peptides (including modify self), and are only activated upon MHC presentation of foreign epitopes.

Minor comments

#1 The following words or expressions should be avoided.

  • “of course” (lane 39-40)

This has been replaced with albeit and reads:

These differences between open and closed ends of the cleft changes the preferred length of the bound peptide, whereby MHC-I often binds shorter (8-10 residues) peptides than MHC-II (>11 residues), albeit with some exceptions 1.

  • “now” (lane 71)

This has been replaced with albeit and reads:

The addition of these recent structures allow us to observe how TCRs can recognise HLA-A*01:01 9, HLA-A*11:01 10, HLA-B*07:02 11, HLA-B*37:01 12, HLA-A*02:06 13, and for the first time, HLA-C 14.

#2 Figures and table 1 are not publication quality.

The figures are good quality on our version, and table 1 has been replaced.

#3 Lane 53 - (D) and (J) definitions are not correct. (D) and (J) stand for Diversity and Joining, respectively.

Apology for the mistake, this has been changed

Diversity in the TCR repertoire is achieved through the random genetic rearrangement of Variable (V or TRV), Diversity (D) and Joining (J) gene segments for TCR β chain (V and J for TCR α chain) 4.

#4 Please explain what the letter omega (Ω) mean in Figure 1 and in the corresponding legend and text? Also precise the abbreviation P for “position” when referring to peptides.

This has been clarified and  the legend of Figure 1 has been changed accordingly:

(A-B) Cleft of MHC-I molecule (pale pink) represented as surface from a top-down view (A) and side view (B), the peptide is represented as pink spheres with the anchors residues at position 2 (P2) and at the last position (PΩ) in the B and F pockets (B).

#5 Mistakes to be corrected in the legend of table 1 (page 4): “chain” instead of “chian”. Also the reference 124 does not appear in the reference list?

Table 1 legend has been updated, and the reference list as well.

#6 Fig.3D - the legend should be changed to reflect the abbreviations shown in the figure: In the legend, P10-Met and P9-Met are mentioned while in the figure one could notice a different abbreviation P10-Metfree and P10-Metbound.

We have changed the figure legend to make it clearer:

The DMF5 TCR binding leads to a register shift of the MMW peptide, whereby the P10-Metfree in the HLA cleft (cyan sticks) is flipped out of the cleft (P10-Metbound in purple stick) upon binding of the DMF5 TCR.

#7 Explain why amino-acid abbreviations differ throughout the text. The authors use “one letter symbol” (30DY31, 98RS99, 4GPD6) or “three-letter” abbreviation (Asp30b, Tyr31b, P4-Pro, P5-Met, P8-Thr etc….). Consistency would be helpful.

The one letter symbol such as in the case of 30DY31, 98RS99, and 4GPD6 is use to quickly denote amino acid motifs that span multiple amino acids within a particular peptide, protein or CDR3 loop sequence. For e.g the one letter symbol is also used to quickly communicate 9mer and 10mer peptide sequences used consistently throughout the paper. Conversely, the three-letter amino acid codes are used to describe single amino acids. The usage of these rules are consistent within our manuscript.

#8 Structured references would be appreciated, in particular for ref.2: the names & surnames of authors are not abbreviated the same way as the other references. Some authors are named without abbreviations (James McCluskey, Mandvi Bharadwaj, etc…), other only with abbreviations (SG, JR). Sometimes “coma” between author names are missing or doubled. Also a ref. including more than 6 authors should include the abbreviation et al. Finally, the year of publication is not 2009 but 2010.

The references have been updated.

Round 2

Reviewer 2 Report

The authors have adequately responded to the comments. The revised version of the manuscript has been significantly improved. No additional amendments are required.